# Demonstrating Biological Fate of Nanoparticle-Loaded Dissolving Microneedles with Aggregation-Caused Quenching Probes: Influence of Application Sites

**DOI:** 10.3390/pharmaceutics15010169

**Published:** 2023-01-03

**Authors:** Yanping Fu, Chaonan Shi, Xiaodie Li, Ting Wen, Qiaoli Wu, Antian Zhang, Ping Hu, Chuanbin Wu, Xin Pan, Zhengwei Huang, Guilan Quan

**Affiliations:** 1College of Pharmacy, Jinan University, Guangzhou 510632, China; 2School of Pharmaceutical Sciences, Sun Yat-sen University, Guangzhou 510006, China; 3The Fourth Affiliated Hospital of Guangzhou Medical University, Guangzhou 511300, China

**Keywords:** dissolving microneedles, solid lipid nanoparticles, in vivo fate, transdermal drug delivery, aggregation-caused quenching

## Abstract

Integrating dissolving microneedles (DMNs) and nanocarriers (NC) holds great potential in transdermal drug delivery because it can simultaneously overcome the stratum corneum barrier and achieve efficient and controlled drug delivery. However, different skin sites with different thicknesses and compositions can affect the transdermal diffusion of NC-loaded DMNs. There are few reports on the biological fate (especially transdermal diffusion) of NC-loaded DMNs, and inaccurate bioimaging information of intact NC limits the accurate understanding of the in vivo fate of NC-loaded DMNs. The aggregation-caused quenching (ACQ) probes P4 emitted intense fluorescence signals in intact NC while quenched after the degradation of NC, had been demonstrated the feasibility of label intact NC. In this study, P4 was loaded in solid lipid nanoparticles (SLNs), and further encapsulated into DMNs, to track the transdermal diffusion of SLNs delivered at different skin sites. The results showed that SLNs had excellent stability after being loaded into DMNs with no significant changes in morphology and fluorescence properties. The in vivo live and ex vivo imaging showed that the transdermal diffusion rate of NC-loaded DMNs was positively correlated with skin thickness, with the order ear > abdomen > back. In conclusion, this study confirmed the site-dependency of transdermal diffusion in NC-loaded DMNs.

## 1. Introduction

Transdermal drug delivery (TDD) is an administration route through the skin to achieve local or systemic therapeutic purposes [1]. It can avoid the metabolism of drugs by gastrointestinal digestive enzymes and overcome the first-pass effect of the liver [2,3]. However, the hindrance of the cutaneous stratum corneum reduces the efficiency of TDD, limiting its use in current clinical practice [4]. To overcome the stratum corneum barrier and improve the transdermal penetration of drugs, various advanced strategies (such as ultrasound, light, heat, and electroporation) have been developed [5]. These approaches, however, have still shown limited efficiency for the transdermal delivery of macromolecular drugs [6].

In recent years, microneedles (MNs) have drawn widespread attention as a novel physical permeation promotion technique. MNs consists of hundreds of tiny micron-sized needle tips attached to a base in an array pattern that can break through the stratum corneum barrier to produce multiple mechanical pores. Thus, MNs can significantly improve the efficiency of transdermal drug delivery [7]. As an intensively studied category of MNs, dissolving microneedles (DMNs) are mainly prepared by water-soluble polymers, and the drugs are distributed in the matrix of the needle tip. The needle tip can dissolve by absorbing tissue fluid after piercing the skin, and the encapsulated drugs are released [8]. DMNs have been widely explored in TDD based on the significant advantages of simple preparation methods, high drug delivery efficiency, and satisfactory patient compliance [9,10,11,12].

Nevertheless, there are still some challenges for monotonic DMNs. Firstly, the hydrophilic material of DMNs severely limits the encapsulation of hydrophobic drugs, while most available drugs in the pipeline are hydrophobic. Secondly, without release management designs, monotonic DMNs cannot achieve precise and controlled drug release [13,14]. Within decades, research based on nanocarriers (NC) in drug delivery has gained wide attention [15]. Various types of NC, such as liposomes [16], solid lipid nanoparticles (SLNs) [17], metal-organic frameworks [18], and magnetic nanoparticles [19], have shown excellent therapeutic effects. NC are capable of loading insoluble drugs and improving bioavailability in vivo [20]. In addition, the modulation of the physicochemical properties of NC enables controlled drug release in a spatiotemporal manner [21]. Therefore, the combination of NC and DMNs can simultaneously overcome the stratum corneum barrier and achieve efficient and controlled drug delivery, producing a synergistic effect.

However, to date, most studies based on NC-loaded DMNs have refrained from preclinical investigation, and it is rare to identify NC-loaded DMNs that have been introduced into the clinic or market. This is mainly attributed to the fact that little is known about the biological fate of NC-loaded DMNs.

Previously, our group preliminarily explored the effect of the length of DMNs on the in vivo fate of NC in both the temporal and spatial dimensions [22]. Noticeably, without a consensus on the standard operating procedure or clinical practice of MNs administration, the application sites of NC-loaded DMNs can be diverse, and NC will be exposed to different biological environments. Differences in skin structures and compositions at different sites [23] may affect the transdermal diffusion rate of NC-loaded DMNs. Therefore, the fate of NC-loaded DMNs depends on the encountered in vivo biological environment [24]. When studying the biological fate of NC, specific physiological conditions must be taken into account. Unfortunately, there are few reports on how different skin sites affect the biological fate of NC-loaded DMNs, which has limited their clinical translation. Even worse, in previous studies, the lack of effective tools and strategies for reliably identifying intact NC has resulted in inaccurate bioimaging information. One of the key issues in the study on the in vivo fate of NC-loaded DMNs is the integrity of NC, which is a prerequisite for improved drug targeting and local retention [25]. Thus, it is impossible to grasp a comprehensive and accurate understanding of the in vivo fate of NC-loaded DMNs.

To demonstrate the biological fate of NC-loaded DMNs administrated via different skin sites, a platform for in vivo comprehensive identification and real-time, precise monitoring of intact NC is essential. Recently, environmentally responsive fluorescent probes have been used to study the in vivo fate of NC [26,27,28,29,30]. The aggregation-caused quenching (ACQ) probes are environment-responsive fluorescent probes with strong hydrophobicity [31]. When ACQ probes are embedded in intact NC, they emit a fluorescent signal. When probes are released as the NC degrade, they tend to aggregate and quench spontaneously due to the ACQ effect [31,32]. Therefore, ACQ probes can identify intact NC.

In this study, we chose SLNs as model NC and further prepared them into NC-loaded DMNs, whose biological fate was scrutinized. Due to the simple production process, high biocompatibility, and excellent drug-loading ability, SLNs are promising for clinical translation [33]. In addition, SLNs are lipid-based NC with a core-shell structure: a hydrophilic surfactant shell and a hydrophobic lipid core, which is capable of effective loading of hydrophobic ACQ probes. Importantly, a series of studies have been conducted on SLNs-loaded DMNs, which showed good compatibility of SLNs with DMNs [22,28,34]. As shown in Figure 1, we first loaded ACQ probes (code P4) into SLNs (P4 SLNs), which were further encapsulated into DMNs (P4 SLNs@DMNs). Then, P4 SLNs@DMNs were applied to the skin of the ear, back, and abdomen in rats, respectively. Subsequently, the biological fate of P4 SLNs@DMNs administrated via different skin sites was monitored by in vivo and ex vivo fluorescence bioimaging. This study is expected to clarify the relationship between the administration site and the in vivo fate of NC-loaded DMNs and provide a theoretical basis for their clinical application.

## 2. Materials and Methods

### 2.1. Materials

The P4 probes were donated by Prof. Wei Wu’s group from Fudan University (Shanghai, China). Gelatin, Tween 80 (T80), and Cetyl Palmitate (CP) were purchased from Aladdin Industrial, Inc. (Shanghai, China). Dichloromethane was acquired from Damao Chemical Reagent Factory (Tianjin, China). Polydimethylsiloxane (PDMS, Sylgard 184 Silicone Elastomer Kit) was obtained from Dow Corning Ltd. (Midland, MI, USA). Sodium hyaluronic acid (HA) (MW < 10 kDa) was supplied by Bloomage Freda Biopharm Co., Ltd. (Jinan, China). Polyvinyl pyrrolidone (PVP K90) was kindly provided by MBCHEM Ltd. (Monmouth Junction, NJ, USA).

### 2.2. Animals

SD male rats (180–220 g) were supplied by Guangdong Medical Laboratory Animal Center (Guangzhou, China). During the experiment, rats were housed under standard conditions of 12-h light/dark cycles and with free access to food and water. All experiments were approved by the Laboratory Animals Ethics Committee of Sun Yat-sen University (Guangzhou, China). (Approval No. SYSU-IACUC-2022-001225).

### 2.3. Synthesis of P4 SLNs

According to the previous study [22], P4 SLNs were prepared through an ultrasound approach. Briefly, 315 mg of CP was heated at 70 °C as the oil phase. At the same temperature, 210 mg of Tween 80 was mixed with 10.5 mL of ultrapure water, acting as the water phase. When CP was completely melted, 1 mL of P4 dichloromethane solution (40.4 μg/mL) was pipetted into the oil phase and stirred for 20 min to evaporate dichloromethane. Subsequently, the water phase was slowly added to the oil phase and stirred continuously at 70 °C for 45 min to form the primary emulsion. Then, the crude emulsion was sonicated for 6 min in an ice bath using an Ultrasonic Cell Disrupter (BI-LON-650Y, BILON Co., Ltd., Shanghai, China). P4 SLNs were ultimately produced after 45 min of churning in an ice bath.

### 2.4. Characterization of P4 SLNs

On days 0, 1, 3, 5, 7, and 10 following preparation, the mean particle size, polydispersity index (PDI), and zeta potential of P4 SLNs were examined using Malvern Zetasizer (Nano ZS90, Malvern Instruments Ltd., Worcestershire, UK). The samples were adequately diluted by 121 folds with ultrapure water before measurement. The fluorescence intensity change of P4 SLNs was measured using a fluorescence spectrometer (Fluoromax-4, HORIBA Ltd., Kyoto, Japan) at days 0, 1, 3, 5, 7, and 10 after preparation. The excitation/emission wavelengths of P4 were 620/660 nm, and the slit width was 4 nm. The stability of P4 SLNs was evaluated over ten days during the storage period. Transmission electron microscopy (TEM, JEM-1400Flash, JEOL Ltd., Tokyo, Japan) was used to observe the morphology of P4 SLNs. The samples were stained with phosphotungstic acid (1% *w*/*v*).

### 2.5. ACQ Property Verification of P4

#### 2.5.1. Water-Quenching Sensitivity

To verify the ACQ properties of the P4 probes, fluorescence emission spectra of P4 in water-acetonitrile co-solvent with 10–100% (*v*/*v*) water content were recorded by fluorescence spectrometer.

#### 2.5.2. Fluorescence Quenching in Biological Matrices

P4 SLNs were mixed with a 7-fold volume of phosphate-buffered saline, and homogenate of rat back, abdomen, and ear skin, respectively. The systems were incubated at 37 °C with a gas bath oscillator (TH2-82BA, Runhua Co., Ltd., Xinghua, China). At predetermined time intervals (2, 4, 6, 8, and 24 h), the fluorescence intensity was recorded by fluorescence spectrometer.

### 2.6. Fabrication and Characterization of P4 SLNs@DMNs

#### 2.6.1. Fabrication

The three-step centrifugation method, as described previously [34], was used to prepare P4 SLNs@DMNs. The female mold exhibited a hole depth of 800 μm, and it was filled with P4 SLNs suspension before being centrifuged at 4000× *g* rpm for 5 min at 4 °C and removing the excess P4 SLNs suspension. The female mold was centrifuged for 1 h under the same conditions and dried in a dryer for an additional night at room temperature. This process was carried out three times to increase the loading effectiveness of P4 SLNs in microneedles. Next, HA solution (400 mg/mL) was filled into the female mold under centrifugation at 4000× *g* rpm for 5 min at 4 °C. After removing the excess HA solution, 250 µL of PVP K90 solution (310 mg/mL) was poured, and centrifuged under the same condition to form the base part. Finally, the female mold was dried for 24 h at room temperature in a dryer, and then the P4 SLNs@DMNs were gently peeled off.

#### 2.6.2. Characterization

The distribution of P4 SLNs in DMNs was observed by confocal laser scanning microscopy (CLSM, LSM800, Carl Zeiss, Oberkochen, Germany). To evaluate the in vitro skin insertion ability, P4 SLNs@DMNs were pressed onto the excised back, abdomen, and ear skin obtained from SD rats for 2 min. After the DMNs were removed, the insertion sites were stained with 1% trypan blue and then imaged with a camera. Subsequently, the skin sites inserted by DMNs were fixed in 4% paraformaldehyde, embedded in paraffin, and then stained by hematoxylin and eosin (H&E). To simulate P4 SLNs@DMNs dissolution in vitro, the DMNs were inserted into the gelatin block with 35% water (*w*/*w*), and the morphology of DMNs after 10 min in the gelatin block was observed with a biomicroscope (BX53, OLYMPUS, Tokyo, Japan), and 3D fluorescence images were captured by CLSM. To evaluate the stability of P4 SLNs encapsulated into DMNs over ten days at storage period, P4 SLNs@DMNs were dissolved by ultrapure water at day ten after preparation and were observed by TEM. The fluorescence intensity change of dissolved P4 SLNs@DMNs dispersions was measured using a fluorescence spectrometer at days 0, 1, 3, 5, 7, and 10 after preparation. Triplicate measurements were performed for each sample.

### 2.7. In Vivo Live Imaging

Before the experiment, hair from the back, abdomen, and ear of SD rats was shaved off to avoid autofluorescence during imaging. After P4 SLNs@DMNs were inserted into the back, abdomen, and ear skin of rats, fluorescence was recorded using in vivo imaging system (IVIS, Lumina Series III, PerkinElmer, Waltham, Massachusetts, America) at predetermined time points (0.5, 2, 4, 8, 12, and 24 h) under excitation/emission wavelengths of 640/710 nm. Fluorescence signals were quantified by region of interest (ROI) analysis and normalized for comparison. Pharmacokinetic analysis was then performed to calculate half-life (*T*_1/2_) and area under the curve (AUC_0–t_). The rats were imaged under anesthesia with isoflurane.

### 2.8. Ex Vivo Imaging

The diffusion behavior of P4 SLNs delivered via different skin sites by DMNs was determined by CLSM. After P4 SLNs@DMNs were inserted into the skin of the back, abdomen, and ear of rats for 4 h, the skin of rats in different site administration groups was collected. The collected skin was then analyzed by 3D reconstruction using CLSM at a magnification of 20×. In addition, three rats were sacrificed in each group at 4, 12, and 24 h post-administration, and the major organs, such as the heart, liver, kidney, spleen, and lungs were collected and imaged by the IVIS system.

### 2.9. Statistical Analysis

All data are expressed as the mean ± standard deviation (SD) from multiple independent experiments, and statistical analysis was performed using one-way analysis of variance (ANOVA) via GraphPad Prism (version 8.02, GraphPad Software LLC., San Diego, CA, USA). The *p*-value < 0.05 was considered to be statistically significant.

## 3. Results

### 3.1. Preparation and Characterization of P4 SLNs

P4 SLNs were fabricated through an ultrasound method. The average hydrodynamic diameter of P4 SLNs was approximately 154 nm (Figure 2A). TEM image further demonstrated uniform particle size and spherical shape of P4 SLNs (Figure 2A). During the 10-day storage period, the particle size of P4 SLNs remained essentially unchanged, around about 150 nm, and the PDI was below 0.2 (Figure 2B). The color of the P4 SLNs solution did not change, and no aggregation and sedimentation of nanoparticles were observed (Figure 2C). In addition, the zeta potential and fluorescence intensity of P4 SLNs did not change significantly within ten days (Figure 2D,E). These results indicated the excellent compatibility of P4 probes with SLNs, and fluorescence quenching did not occur for P4 probes loaded in SLNs during storage.

### 3.2. ACQ Property Verification of P4

The photophysical properties and ACQ properties of the P4 probes were investigated to confirm their feasibility in bioimaging (Appendix A). Appendix A show the contour fluorescence spectra and fluorescence emission spectra of P4 probes, respectively. It demonstrated that the maximum excitation and emission wavelength of P4 probes were 640 nm and 660 nm, respectively, which is similar to the previous study [26]. As shown in Appendix A, the fluorescence emission intensities of P4 decreased as the proportion of water content increased. In particular, when the proportion of water reached 90% (*v*/*v*, turning point) or higher, only the baseline signal was detected in the spectrum. These results indicated that the P4 probes underwent complete quenching in the biological environment (e.g., skin tissue fluid) with nearly 100% water content.

To further investigate the quenching behavior of P4 SLNs in skin tissue fluid, P4 SLNs were co-incubated with homogenates from different parts of the rat skin (Figure 2F). The fluorescence intensity of P4 SLNs co-incubated with different skin homogenates significantly reduced compared to PBS. It was possibly due to the degradation of P4 SLNs by enzymes in the skin tissue fluid leading to the subsequent release of P4, which resulted in fluorescence quenching. At 24 h, the fluorescence intensity of the back, abdomen, and ear skin homogenate groups decreased to 35%, 36%, and 45%, respectively. In contrast, the fluorescence intensity of the PBS group showed essentially no decrease. As shown in Appendix A, when the P4 solution was incubated with phosphate-buffered saline solution, the fluorescence signal of the P4 solution group almost completely disappeared at 0.5 h. Hence, the results indicated that SLNs could be degraded by biological matrices such as skin tissue fluids, resulting in the ACQ phenomenon of P4.

### 3.3. Fabrication and Characterization of P4 SLNs@DMNs

In order to improve the loading rate of P4 SLNs in DMNs, this study used multiple centrifugation steps to enrich P4 SLNs at the tip of the needle, as described in our previous study [34] (Figure 3A). The DMNs patch consisted of 144 needles (12 × 12) with a height of 800 μm in the shape of a quadrilateral cone. The fluorescence images of P4 SLNs@DMNs (Figure 3B) captured by CLSM depicted that the P4 SLNs were mainly concentrated in the tips of needles.

To assess the dissolution behavior of P4 SLNs@DMNs, a DMNs patch was inserted into a gelatin block with a similar level of hydration to that of the skin stratum corneum [35] and removed at predetermined time points. The side view of the microscope in Figure 3C shows that the P4 SLN@DMNs could dissolve completely within 10 min. Figure 3D shows the fluorescence image of P4 SLNs remaining in the gelatin block after the removal of the base of DMNs. These results demonstrated that the DMNs were able to penetrate the gelatin block and dissolve rapidly and deliver the P4 SLNs inside the gelatin block.

### 3.4. Insertion Ability of P4 SLNs@DMNs

The photograph after trypan blue staining (Appendix A) and H&E-stained section (Figure 3E) of the skin after insertion with DMNs show that DMNs patch could form microchannels in rat skin. These results demonstrated that DMNs could successfully penetrate the epidermis of different parts of the rat skin, which was the prerequisite to the study of the fate of NC delivered in vivo by DMNs. Taken together, DMNs have been demonstrated to be mechanically strong enough to pierce the skin and subsequently dissolve rapidly through the skin tissue fluid, allowing for effective delivery of P4 SLNs.

### 3.5. Stability of P4 SLNs@DMNs

We investigated the stability of P4 SLNs@DMNs to ensure that the physicochemical properties of P4 SLNs did not change after they were loaded into DMNs. As shown in Appendix A, no significant change in the morphology of P4 SLNs@DMNs was observed over ten days. TEM image showed a slight increase in particle size after P4 SLNs were loaded into DMNs (Appendix A), probably due to HA adhesion to its surface, which did not affect the property of P4 SLNs themselves.

Moreover, the fluorescence intensity of dissolved P4 SLNs@DMNs suspension was measured during the 10-day storage period. As indicated in Figure 3F, the fluorescence intensity of P4 within P4 SLNs@DMNs was still maintained unaltered during storage for ten days. In general, the prepared DMNs exhibited superior stability and would not alter the structure and fluorescence properties of the encapsulated P4 SLNs.

### 3.6. In Vivo Live Imaging

The diffusion rate of P4 SLNs delivered via different skin sites by DMNs was monitored by the IVIS system in rats. Figure 4A shows the live images of rats treated with P4 SLNs@DMNs at different skin sites. In the group without any treatment on the skin of the back, abdomen, and ear, no fluorescent signal associated with the P4 probe was detected (Appendix A). Over time, the fluorescence intensity indicated by the retention of P4 SLNs in different parts of the rat skin gradually decreased (Figure 4B–D), associated with particle diffusion. The results of live imaging implied differences in the retention profile of P4 SLNs in the rats.

The fluorescence intensity measured at different time points for each group was normalized to the fluorescence intensity of the first time point (0.5 h) and assigned an intensity value of 100%. Plotting the fluorescence intensity over time showed a significant decreasing trend for all groups (Figure 4E). After 8 h of administration, the percentage of fluorescence intensity decreased to 83.72 ± 0.10%, 65.07 ± 0.06%, and 41.92 ± 0.04% for back, abdomen, and ear skin, respectively. The rate of decrease in relative fluorescence intensity was in the order of ear > abdomen > back. Furthermore, within 24 h, the AUC_0–t_ values of the back, abdomen, and ear were 1146.40%·h, 952.16%·h, and 560.99%·h, respectively (Figure 4F). The T_1/2_ values of the back, abdomen and ear were calculated to be 11.85 h, 9.39 h, and 7.08 h, respectively (Figure 4G).

### 3.7. Ex Vivo Imaging

To visualize the diffusion rate of P4 SLNs, CLSM was used to observe the diffusion of P4 SLNs after treatment at different sites in rats. As shown in Figure 5A, the group of back remained high fluorescence after 4 h of administration, implying that P4 SLNs maintained a certain degree of integrity. In contrast, the group of ear exhibited weaker fluorescence, indicating that most of the P4 SLNs were degraded after 4 h of administration, resulting in P4 leakage and quenching of fluorescence. Moreover, compared to the group of abdomen and ear, the group of the back could observe the fluorescent signal deepening in the skin after 4 h of administration by DMNs. The retention of fluorescence signal was in the order of back > abdomen > ear. Furthermore, the fluorescent signal of the back could be detected at a deeper level of the skin. The maximum skin depth of fluorescent signals detected in the back, abdomen, and ear were 140 μm, 120 μm, and 100 μm, respectively. In addition, almost no fluorescence signal was detected in major organs such as the heart, liver, spleen, lung, and kidney after 4, 12, and 24 h of administration with P4 SLNs@DMNs (Figure 5B). Moreover, there was still no fluorescent signal of skin and organs observed in the group without any treatment (Appendix A).

## 4. Discussion

In the past few decades, the rapid development of pharmaceutical and material technology has made it possible to combine DMNs with novel NC, and the combination of the two has significantly broadened the application of TDD and demonstrated significant efficacy in various disease models [36,37,38]. However, there are few products of NC-loaded DMNs currently on the market, and they still face many challenges in practical application. First, there are a limited number of studies concerning in vivo fate of NC-loaded DMNs. Moreover, in previous studies [39,40], the fluorescent signals detected were a mixture of NC and free probes, which could not represent intact NC and did not accurately elucidate the in vivo fate of NC. Thus, there is an urgent need for a reliable bioimaging tool for holistic identification and accurate detection of NC to facilitate the clinical translation of NC-loaded DMNs. Second, current studies [26,28,41] on the in vivo fate of NC have focused more on the delivery system itself, ignoring the influence of the complex biological environment. The skin is the largest organ of the body, and the sites of DMNs application may affect the in vivo fate of NC-loaded DMNs, mainly because at different skin sites: (1) Different thicknesses of the epidermis and dermis meant that the area where the needle tips enter the dermis after insertion of DMNs was different; (2) Different classes of cells and matrices in the dermal region implied different interactions with NC; (3) Different mechanical strength meant different shear forces on the DMNs [23]. The above factors might mutually affect the transdermal absorption rate of NC-loaded DMNs, and it was necessary to investigate the role of the site of administration on the fate of NC in vivo. Based on this rationale, we chose P4 probes with ACQ effect as a bioimaging tool for in vivo fate of intact NC. Moreover, we constructed P4 SLNs@DMNs that were applied to the back, abdomen, and ear skin of rats, respectively, to investigate the effects of different skin sites on the in vivo fate of P4 SLNs@DMNs (Figure 1).

Based on the previous study of our group [34], we have prepared SLNs@DMNs systems with excellent biocompatibility and safety. Furthermore, we have demonstrated the storage stability of P4 SLNs through experiments (Figure 2B–E). During the storage period, the particle size, PDI, and fluorescence intensity of P4 SLNs did not change significantly. In addition, the fluorescence quenching occurred when P4 probes loaded into SLNs were exposed to biological matrices (Figure 2F), which not only validated the ACQ properties of P4 probes but also demonstrated the feasibility of P4 probes in bioimaging. Furthermore, we also investigated the physicochemical properties of P4 SLNs@DMNs. The results showed that the morphology of the microneedles was intact, P4 SLN was mainly distributed at the needle tip (Figure 3B,C), and the microneedles had good puncture ability and stability (Figure 3D–F), which ensured the effective transdermal delivery of P4 SLN.

In the in vivo live imaging study (Figure 4A), we found significant differences in the diffusion rates of SLNs loaded in DMNs applied at different sites, and the order of the rate of diffusion was as follows: ear > abdomen > back. The application of P4 SLNs@DMNs to the back resulted in higher AUC_0–t_ and *T*_1/2_, while the application to the ear resulted in the lowest AUC_0–t_ and *T*_1/2_ (Figure 4F,G), which is consistent with the previous findings [42]. A reasonable explanation is that ear has the thinnest thickness of the stratum corneum, the least number of cell layers, and the lowest elastic properties compared to the back and abdomen [43,44], and thus SLNs are more likely to enter the dermal region of the ear. Moreover, the capillaries are most abundant in the dermal region of the ear compared to the back and abdomen [45], and therefore, SLNs are more likely to enter the body’s circulation through the capillaries. In the ex vivo imaging study of rat skin (Figure 5A), we found that the most fluorescence remained in the back skin after 4 h of application with P4 SLNs@DMNs, while the fluorescence signal in the ear almost disappeared. This is similar to the trend of fluorescence reduction in the in vivo live imaging study. These results suggest that SLNs are more likely to accumulate in thicker skin areas.

In addition, no strong fluorescent signal was detected in the major organs after applying P4 SLNs@DMNs to different skin sites (Figure 5B), and there was almost no distribution of integrated SLNs in the major organs. It seems to indicate that most NC are degraded after entering the body circulation via DMNs administration, which further suggests that the target-modified ligands may be lost, and perhaps off-target effects may occur after the administration of target-modified NCs-loaded DMNs. Therefore, more attention should be paid when designing NCs-loaded DMNs.

In general, our results have implications for the selection of transdermal delivery sites for the therapy of different diseases (Figure 6). For example, for chronic diseases in which patients require long-term medication, such as hypertension, diabetes, rheumatoid arthritis, etc., perhaps the application of NC-loaded DMNs via the thicker skin areas of the back and forearm is an option to consider. Instead, for acute diseases, in which the drug needs to work quickly, such as motion sickness, coronary heart disease, acute pains, etc., the skin of the posterior auricle and forehead, which are rich in capillaries, is ideal [46,47]. Overall, the difference in NC-loaded DMNs diffusion rates observed at different sites of the skin can provide a theoretical basis for the design of selective therapies. What is more, it indicates the importance of clearly defining the site of administration for different indications during the formulation development process, and relevant standard operating procedures or clinical practices should be put forward, and then the therapeutic effect could be maximized. We also suggest that when the product of NC-loaded DMNs is launched in the future, the site of administration could be specified in the instructions in order to improve the treatment effect of patients.

In conclusion, different skin sites are important factors influencing the fate of transdermal delivery of NC-loaded DMNs. Apart from differences between sites, skin varies in different disease conditions (such as psoriasis, proliferative scarring, and subcutaneous tumors) and even in people of different ages and genders [23,48,49]. Therefore, the in vivo fate of NCs loaded in DMNs needs to be further investigated in such different cases. Furthermore, our study mainly analyzed the transdermal delivery efficiency of NC-loaded DMNs at different skin sites from a macroscopic perspective. We will expand our study to explore the interactions between NC-loaded DMNs and different skin tissue cells from a microscopic perspective in ongoing studies.

## 5. Conclusions

In this study, P4 SLNs@DMNs with good stability was successfully constructed, which exhibited the good compatibility of SLNs with DMNs. Moreover, DMNs could effectively pierce different sites of rat skin, followed by the rapid dissolution and release of the loaded SLNs. The results of in vivo live imaging and ex vivo imaging showed that the order of the rate of diffusion was as follows: ear > abdomen > back, and NCs were more likely to accumulate in the back skin. Therefore, the appropriate site of drug delivery can be selected according to the rate and maintenance time of drug onset required for different diseases. These findings can provide a strong theoretical basis for the clinical application of NC-loaded DMNs. In addition, the effects of skin in different disease states, different ages, and genders on the in vivo fate of NC-loaded DMNs should be further explored to promote clinical translation.

## Figures and Tables

**Figure 1 pharmaceutics-15-00169-f001:**
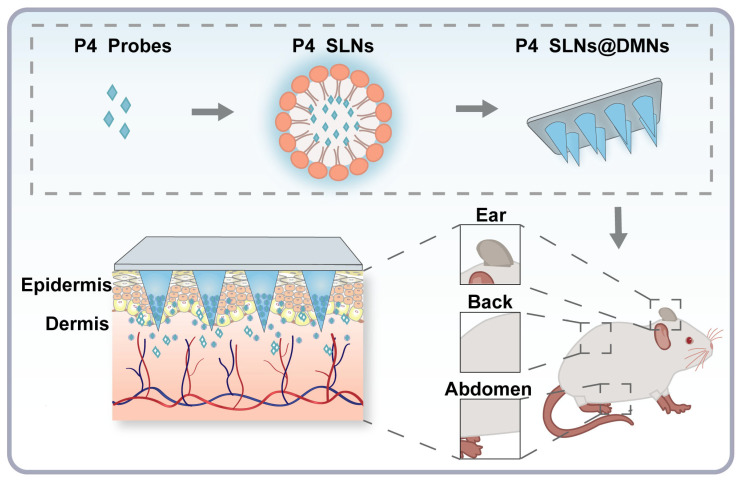
Schematic illustration of the research framework: Impact of different skin sites on in vivo fate of P4 SLNs@DMNs.

**Figure 2 pharmaceutics-15-00169-f002:**
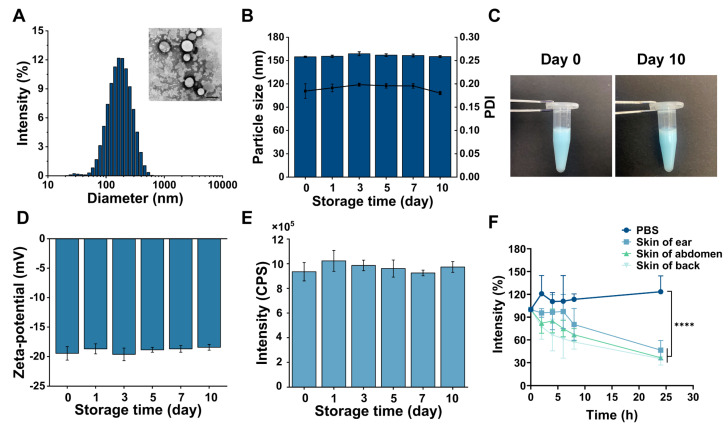
Preparation and characterization of P4 SLNs. (**A**) Particle size distribution and TEM image of P4 SLNs (scale bar: 200 nm). (**B**) Stability of the averaged particle size and PDI of P4 SLNs (*n* = 3). (**C**) The appearance of P4 SLNs before and after ten days. (**D**) Stability of zeta potential of P4 SLNs (*n* = 3). (**E**) Peak values of emission spectra of P4 SLNs over ten days (*n* = 3). (**F**) Fluorescence quenching of P4 SLNs in biological matrices (*n* = 3). Data are expressed as mean ± SD. Note: **** denotes *p* < 0.0001 vs. group of PBS.

**Figure 3 pharmaceutics-15-00169-f003:**
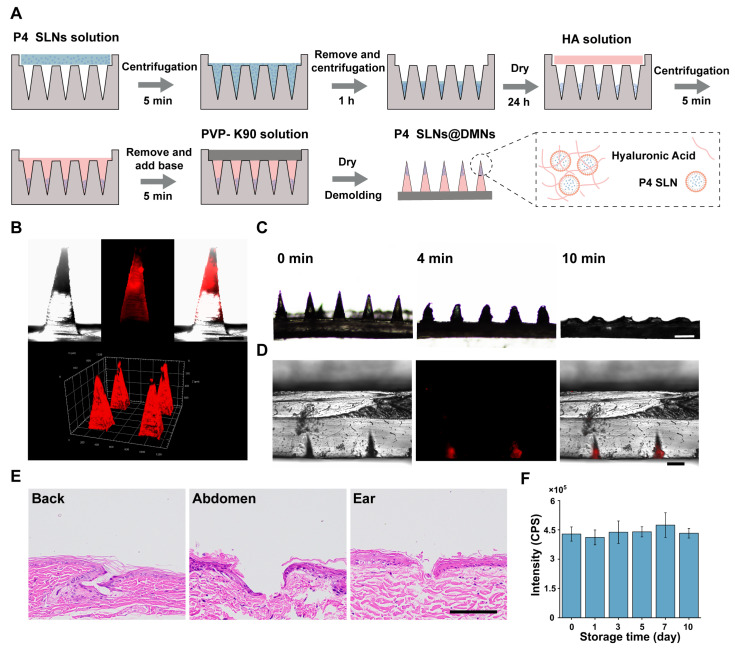
Preparation and characterization of P4 SLNs@DMNs. (**A**) Scheme of preparation process of P4 SLNs@DMNs. (**B**) The fluorescence images of P4 SLNs@DMNs (scale bar: 200 μm). (**C**) Optical microscopy images of P4 SLN@DMNs before and after insertion into gelatin (scale bar: 500 μm). (**D**) The fluorescence images of gelatin block after insertion of P4 SLNs@DMNs for 10 min (scale bar: 200 μm). (**E**) The histological section of back, abdomen, and ear skin of rats after P4 SLNs@DMNs application (scale bar: 100 μm) (**F**) Peak values of emission spectra of dissolved P4 SLNs@DMNs dispersions within ten days (*n* = 3). Data are expressed as mean ± SD.

**Figure 4 pharmaceutics-15-00169-f004:**
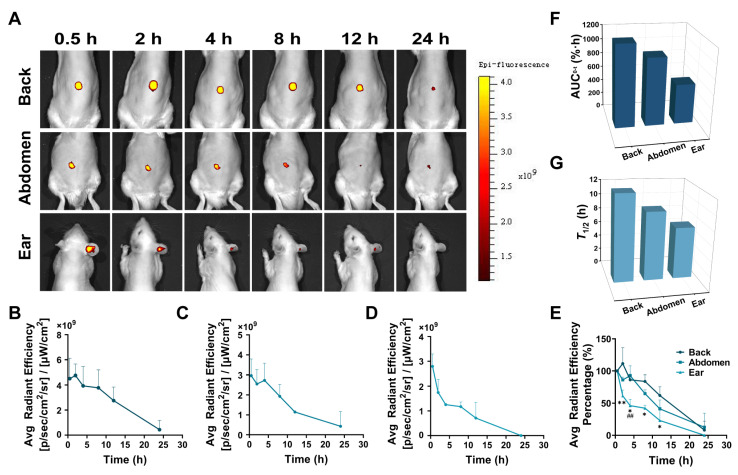
In vivo live imaging. (**A**) Representative live images of rats treated with P4 SLNs@DMNs at the skin of the back, abdomen, and ear. (**B**–**D**) Average fluorescence intensity of rats treated with P4 SLNs@DMNs at the skin of the back, abdomen, and ear (*n* = 3). Data are expressed as mean ± SD. (**E**) The relative fluorescent intensity of rats treated with P4 SLNs@DMNs (*n* = 3). Data are expressed as mean ± SD. Note: ** denotes *p* < 0.01 vs. group of back, * denotes *p* < 0.05 vs. group of back. **##** denotes *p* < 0.01 vs. group of abdomen. (**F**) The AUC_0–t_ values and (**G**) *T*_1/2_ values of relative fluorescent intensity of rats treated with P4 SLNs@DMNs at the skin of the back, abdomen, and ear.

**Figure 5 pharmaceutics-15-00169-f005:**
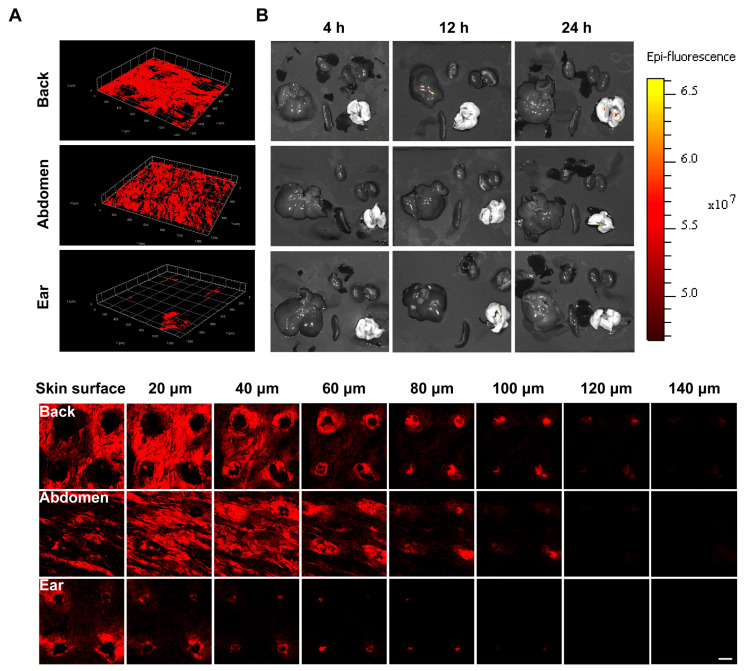
Ex vivo imaging. (**A**) Representative CLSM 3D reconstruction images (up) and 2D fluorescent images (down) of the skin of the back, abdomen, and ear of rats after being treated with P4 SLNs@DMNs at 4 h (scale bar: 100 μm). (**B**) Representative fluorescent images of major organs after being treated with P4 SLNs@DMNs at 4, 12, and 24 h.

**Figure 6 pharmaceutics-15-00169-f006:**
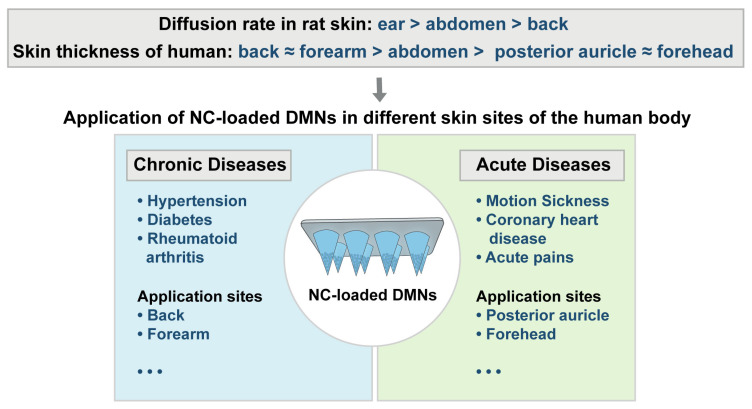
Potential application of NC-loaded DMNs at different skin sites.

## Data Availability

Not applicable.

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
