# Peer review of "Demonstrating Biological Fate of Nanoparticle-Loaded Dissolving Microneedles with Aggregation-Caused Quenching Probes: Influence of Application Sites"

_pharmaceutics, 2023, doi:10.3390/pharmaceutics15010169_

Round 1

Author Response

We sincerely thank the reviewer's constructive comments. We have responded to the reviewer's comments point-by-point, and we have carefully revised the manuscript. Please see the attachment.

Reviewer 2 Report

The manuscript by Fu et al., titled “Demonstrating biological fate of nanoparticle-loaded dissolving microneedles with aggregation-caused quenching probes: Influence of application sites” is describing the impact of different skin sites on the transdermal diffusion of nanocarriers-loaded into integration of dissolving microneedles (NC-loaded DMNs). The authors claimed that the results showed different transdermal diffusion rates at different skin sites. The current work is important and provides new insights about choosing proper administration sites for microneedles. Overall manuscript is well written. Therefore, I recommend publication in Pharmaceutics after addressing the following points.

-          Some results lack statistical analysis. Please consider statistical analysis for figures 2f and figure 4e.

-          Nothing was mentioned about the toxicity of the prepared nanoparticles and DMNs. Please comment if they are safe to use.

Author Response

(The authors gave the same response as above.)

Round 2

Reviewer 1 Report

I found the authors have addressed all the queries raised by the reviewer. Now it can be accepted for publication in the present format.